# Insulin-Loaded Barium Salt Particles Facilitate Oral Delivery of Insulin in Diabetic Rats

**DOI:** 10.3390/pharmaceutics12080710

**Published:** 2020-07-29

**Authors:** Rahela Zaman, Md. Emranul Karim, Iekhsan Othman, Anuar Zaini, Ezharul Hoque Chowdhury

**Affiliations:** Jeffrey Cheah School of Medicine and Health Sciences, Monash University, Subang Jaya 47500, Malaysia; rahela.zaman@monash.edu (R.Z.); karim604306@gmail.com (M.E.K.); iekhsan.othman@monash.edu (I.O.); anuar.zaini@monash.edu (A.Z.)

**Keywords:** insulin, oral delivery, barium salt particles, barium phosphate, barium carbonate, barium sulfite, acidic pH, pepsin, mucin, sustained release, hyperglycemia, diabetes

## Abstract

Oral delivery is considered as the most preferred and yet most challenging mode of drug administration; especially a fragile and sensitive peptide like insulin that shows extremely low bioavailability through the gastro-intestinal (GIT) route. To address this problem, we have designed a novel drug delivery system (DDS) using precipitation-induced Barium (Ba) salt particles. The DDS can load insulin molecules and transport them through the GIT route. There were several in vitro simulation tests carried out to prove the efficiency of Ba salt particles as oral delivery candidates. All three Ba salt particles (BaSO_4_, BaSO_3,_ and BaCO_3_) showed very good loading of insulin (>70% in all formulations) and a degree of resistance throughout a wide range of pHs from basic to acidic conditions when assessed by spectrophotometry. Particles and insulin-associated particles were morphologically assessed and characterized using FE-SEM and FT-IR. A set of tests were designed and carried out with mucin to predict whether the particles are potentially capable of overcoming one of the barriers for crossing intestinal epithelium. The mucin binding experiment demonstrated 60–100% of mucin adhesion to the three different particles. FT-IR identifies the characteristic peaks for mucin protein, particles, and particle-mucin complex re-confirming mucin adhesion to the particles. Finally, the effectiveness of nano-insulin was tested on streptozotocin (STZ) induced diabetic rats. A short acting human insulin analog, insulin aspart, was loaded into Ba salt particles at a dose of 100 IU/Kg prior to oral administration. Among the three formulations, insulin aspart-loaded BaSO_4_ and BaCO_3_ particles dramatically reduced the existing hyperglycemia. BaSO_4_ with loaded Insulin showed an onset of glucose-lowering action within 1 hr, with blood glucose level measured significantly lower compared to the 2nd and 3rd h (*p* < 0.05). Insulin-loaded BaCO_3_ particles showed a significant decrease in blood glucose level at 1–2 h, although the glucose level started to show a slight rise at 3rd h and by 4th h, it was back to baseline level. However, although BaSO_3_ particles with loaded insulin showed a trend of reduction in blood glucose level, the reduction was not found to be significant (*p* < 0.05) at any point in time. Therefore, oral formulations of insulin/BaSO_4_ and insulin/BaCO_3_ particles were observed as effective as native insulin aspart subcutaneous formulation in terms of onset and duration of action. Further investigation will be needed to reveal bioavailability and mechanism of action of this novel Nano-Insulin formulations.

## 1. Introduction

The emergence of protein therapeutics is being considered as one of the biggest milestones of 20th century medical and pharmaceutical science [1]. Advancement in biotechnology and a better understanding of the pathophysiology of clinical conditions helped several protein and peptide drugs to be recognized as potential therapeutics [2]. The trend is ongoing and in the next few decades, protein and peptide therapeutics will be covering an even bigger share of pharmaceutical products in industry. However, one disturbing fact for protein and peptide therapeutics is that there is a scarcity in noninvasive oral modes of administration. Proteins are naturally hydrophilic molecules with poor absorbance through the intestine and often prone to harsh pH and enzymatic degradation [2]. For example, the first-ever protein molecule used as therapeutic was insulin which, since that time, has been the one and only treatment for type1 diabetes as well as for type2 diabetes mellitus [1]. Yet, the only form of treatment available for insulin is the injectable form. Regular and very frequent subcutaneous (SC) administrations are associated with low patient compliance and multiple injection site injuries. Unfortunately, it is not possible to administrate insulin as an oral tablet. Insulin is a small 51 amino acid peptide molecule that is not resistant to stomach acid or enzymatic degradation.

Delivery through the oral route is generally recognized as the delivery of drugs through the GIT where the start point for a drug can be mouth or buccal and the absorption can take place from different parts of the small and large intestine [3]. Popular searches for oral insulin are mostly restricted to direct delivery from mouth to absorption through the small intestinal wall [3,4,5,6,7,8,9,10]. The complex GI tract is one single tube intended to serve one single purpose of entry and absorbance of nutrients from an outside source and making them available to the body. The tube is divided into different organs with each organ having its own pH and enzymatic environment. For example, stomach and small intestine pH and secretary enzymes are specialized in modifying and breaking peptide and protein molecules. In this process, these organs destroy any protein drug that goes through them non-discriminatorily [3,11]. The pH and enzymes are considered a primary barrier for oral delivery of insulin and other protein drugs. Another barrier is intestinal absorption of high molecular weight protein in intact form. The intestinal wall is covered with a mucus layer which is a combination of glycoproteins and bicarbonate ions. Beneath the mucus is the epithelium, basement membrane (non-cellular layer), and a layer of submucosa that holds blood vessels and lymphatic ducts. There are different modes of uptake of molecules (phagocytosis, absorption through payer’s patches, or endocytosis by enterocytes) depending on the size, surface charge, hydrophilic or hydrophobic nature of the molecule [3,12]. Absorption of an intact protein fails mostly due to its size and hydrophilic nature.

Despite all those setbacks, there have been several strategies proposed to design an oral protein formulation. The concept of attaching a protein molecule to another molecule that envelops and creates a cargo for it is a popular one. Nanoparticle-based strategies are best served for this concept. To be an ideal candidate for oral peptide delivery, a carrier molecule must meet few criteria, such as the ability to bind to a therapeutic molecule, showing adequate resistance to stand stomach harsh pH and enzymatic degradation and capacity to be absorbed through the intestinal wall. Due to the complexity of the task, until now the success rate in designing an oral protein formulation is still very low [3]. A popular choice for oral insulin carriers so far included polymeric nanoparticles, like poly(lactide-co-glycolide) (PLGA) and its derivatives. Insulin incorporated into a blended polymer of poly fumaric anhydride (FA) and PLGA 50: 50 (FA: PLGA) showed a reduction of glucose load in fasted rats for 3 h [4,5]. Another formulation of HP55-coated capsule containing PLGA/RS nanoparticle- loaded insulin gradually lowered glucose level from 10–15 h [6]. Apart from PLGA particles, there were few successful oral nano-insulin delivery tools reported using chitosan (CS) and its derivatives. PH-sensitive CS particles showed a promising 15 h long effect on blood glucose levels in diabetic rats, however, the effect started with a burst release [7]. Similarly, TBA (thiolated polymer 2-Iminothiolane)-attached CS with incorporated insulin showed a glucose-lowering effect for 24 h in non-diabetic rats [13]. Lysosome-entrapped insulin (LEI) also reported for successful oral insulin formulation [14,15]. Despite the success of the animal model, none of the formulations were qualified for the clinical trial. Most recently, oral insulin 338(1338) has been taken out of the Phase II clinical trial, although the formulation showed a good hyperglycemic grip in patients. Oral insulin 338(1338) was created by a combination of technologies involving amino acid substitution and linking to a carrier molecule. Acylation at 1338 via 18 C fatty acid let the insulin to bind reversibly with albumin which extended its half-life in circulation, while attachment to sodium caprate enhanced the absorption. The clinical trial was discontinued based on the finding that insulin 338(1338) was effective in a very high dose which would make production cost high [16]. However, insulin 338(1338) is the formulation that almost made its way to the market, with the most advanced research made in the oral insulin field to date.

Another formulation is insulin tregopil (Biocon), a human insulin analog attached to a methoxy-triethylene-glycol-propionyl moiety linked to the Lys-β29 amino group and formulated with sodium caporate. Administration of 10–30-mg tablets showed promising effects on hyperglycemia. The clinical trial is ongoing and we can learn better about it in a few years. Again, the bioavailability estimated in dogs seems not very high (0.82–0.85%) which is common for oral insulin tablets [17,18]. ORMD-0801 from Oramed was successful in the Phase1 clinical trial and currently in the Phase2 clinical trial. The formulation was prepared by attaching insulin to a permeation enhancer, soybean trypsin inhibitor, and a chelator. Pre-clinical stage bioavailability in animal study was estimated at 5–8% [17,19]. The clinical trial showed a promising effect on controlling blood sugar levels without any noticeable adverse side-effect. Another formulation, Diasome, which can be given both orally and subcutaneously, is liver-targeted. The oral dosing is less effective compared to SC administration [17,20].

The first precipitation-based particle reported for insulin loading for oral delivery was strontium- substituted carbonate apatite which showed limited success in terms of binding and release in in vitro medium [21]. Here we report a novel group of precipitation-based salt particle formulations designed in our lab, having enormous potential as cargo for oral macromolecular drug delivery. There is no harsh temperature or pH processing required during the fabrication process that can contribute to preventing protein degradation or inactivation. In our earlier studies, we reported Group 2 metal sulfate, sulfite, and carbonate as potential delivery vehicles for genetic material and micromolecular drugs to tumor tissue [22,23]. These salt particles are less immunogenic compared to viral vectors and thus considered safer. Moreover, the flexible synthesis conditions make them easily modifiable depending on the target binding molecule and mode of delivery. They have a better drug loading capacity, and the drug release profile can be controlled as well [22]. In this study, we aimed to investigate their effectiveness as delivery vehicles for macromolecular peptides orally. Ba salt particles (BaSO_4_, BaSO_3,_ and BaCO_3_) have both positive and negative charged domains conferred, respectively, by cations (Ba^2+^) and anions (SO_4_^2−^, SO_3_^2−^, and CO_3_^2−^). Due to heterogeneous charge distribution, these particles are capable of binding peptide molecules like insulin through electrostatic interactions. The particles were tested in vitro with excellent protein loading efficiency and release profile. They also demonstrated high resistance to withstand gastrointestinal pH and enzymatic environment along with the ability to efficiently bind mucin, indicating their potential capacity for intestinal absorption. Finally, using a short-acting human insulin analog (insulin aspart), the effectiveness of oral insulin delivery was evaluated for the three different particles in diabetic rats. Among the three candidates, BaSO_4_ and BaCO_3_ particles were found to dramatically reduce the existing hyperglycemia, apparently working as the native insulin Aspart subcutaneous formulation in terms of onset and duration of the action.

## 2. Methodology

### 2.1. Materials and Methods

#### 2.1.1. Reagents

Dulbecco’s Modified Eagle’s Medium (DMEM) powder was purchased from Invitrogen (Waltham, MA, USA). Hydrochloric acid (HCl) (1M), sodium hydrogen carbonate (NaHCO_3_), barium chloride dehydrate (BaCl_2_, 2H_2_O), sodium sulfite (Na_2_SO_3_), and mucin were bought from Sigma Aldrich. Additionally, fluorescein isothiocyanate (FITC)-labelled insulin stock (Human, recombinant, expressed in yeast, lyophilized powder) was purchased from Sigma Aldrich (St. Louis, MO, USA). HEPES (2-(4-(2-hydroxyethyl)-1-piperazinyl)ethanesulfonic acid) and sodium carbonate (Na_2_CO_3_) were from Fisher Scientific (Waltham, MA, USA) and sodium sulfate (Na_2_SO_4_) from Merck (Kenilworth, NJ, USA). Insulin aspart (NovoRapid^®^, Novo Nordisk, Bagsværd, Denmark) was purchased from a local pharmacy. Pepsin was purchased from Promega (Madison, WI, USA).

#### 2.1.2. Reagent Preparation

1M of BaCl_2,_ Na_2_SO_4_, Na_2_SO_3,_ and Na_2_CO_3_ stock solutions were prepared by calculating the amount of the respective molecular weight of powder and dissolving them in water. All solutions were stored in 1 mL aliquots at −20 °C. 100 mL of bicarbonated DMEM solution was freshly prepared by dissolving 1.35 g of DMEM powder in 95 mL pure Milli-Q water, followed by the addition of 0.37 g of sodium hydrogen carbonate (0.44 mM final concentration). The pH of the solution was then adjusted to the desired level by the addition of either 1 M HCl or 1 M NaOH. The final volume was then adjusted to 100 mL. Two mg/mL of FITC-insulin stock solution was prepared by dissolving the FITC-insulin powder into an appropriate volume of pure Milli-Q^®^ water.

### 2.2. Synthesis of Empty and Insulin-Loaded Ba Salt Particles

#### 2.2.1. Synthesis of Particles by Precipitation Reaction

Each type of Ba salt particle (BaSO_4_, BaSO_3_, or BaCO_3_) was prepared by incorporating a volume of cation-providing BaCl_2_ salt into either a volume of HEPES-buffered solution (pH adjusted to 8.0) or milliQH_2_O and mixing the resultant solution with a volume of one anion-providing salt, Na_2_SO_4_, Na_2_SO_3_ or Na_2_CO_3_, respectively. The final mixture was incubated for 30 min at 37 °C and subsequently added to miliQH_2_O to obtain the final volume of particle suspension. Table 1 shows the various concentrations of reacting salts used to fabricate three different Ba salt particles. Figure 1 is a diagrammatic representation of precipitation reaction.

#### 2.2.2. Insulin Loading into Ba Salt Particles

Each type of Ba salt particle was prepared by incorporating a volume of cation- providing BaCl_2_ salt into either a volume of HEPES-buffered solution (pH adjusted to 8.0) or miliQH_2_O and mixing the solution with a volume of one anion-providing salt, Na_2_SO_4_, Na_2_SO_3,_ and Na_2_CO_3_. Insulin was introduced right before addition of the 2nd salt to the medium. The final mixture was incubated for 30 min at 37 °C and subsequently added to milliQH_2_O to obtain the final volume of particle suspension. Table 2 below shows the various concentrations of reacting salts and insulin used for the preparation of insulin-loaded Ba salt particles. Figure 2 shows a simplified diagrammatic depiction of insulin loading into particles.

### 2.3. Characterization and Morphological Screening of Ba Salt Particles

#### 2.3.1. Fourier Transform Infrared Spectroscopy (FT-IR)

A volume of 250 µL of 1M of BaCl_2_ was taken in a centrifuge tube_,_ followed by the addition of 100 µL of 1M Na_2_SO_4_ or Na_2_SO_3_ or Na_2_CO_3_ to the respective tubes to prepare BaSO_4_, BaSO_3_, or Ba_2_CO_3_ particles. The samples were then incubated at 37 °C for 30 min. The final volume of the particle solution was made to 50 mL with the addition of milliQH_2_O. Samples were then centrifuged for 15 min at RPM 5000. Precipitates were separated from supernatant, stored sequentially at −20 °C overnight, and at −80 °C for 0.5 h and finally placed into a freeze-dryer (Labconco freeze dryer, Kansas City, MO, USA) for 5 h. Samples were read using a Varian FT-IR (Santa Clara, CA, USA) and analyzed with Resolution Pro 640 software.

#### 2.3.2. Elemental Analysis of Particles by Energy Dispersive X-Ray Spectroscopy (EDX)

5 µL of cation- providing salt was added to 20 µL milliQH_2_O, followed by the addition of 2 µL of anion-providing salt. After 30 min of incubation at 37 °C, 1 mL of Milli-Q H_2_O was added. 2 µL of the solution containing the particles was transferred on a coverslip. The coverslip was then air-dried in elevated temperature (45 °C) inside a dryer for 1 h. The dried sample was marked with a circular line with a marker for easy identification under a microscope. Samples were then subjected to platinum sputtering for 45 s with 30 mA and a factor of 2.3. Visualizations and documentation were done using a Field Emission Scanning Electron Microscope (FE-SEM) (Hitachi/SU8010, Tokyo, Japan) at 5.0 kv.

### 2.4. Particle Stability Assessment

Ba salt particles were prepared as stated in the particle preparation Section 2.2.1. After 30 min of incubation at 37 °C, 1 mL of DMEM of different pHs (pH 7.78 to pH 1.78) was added and kept at 37 °C. Samples were prepared in triplicates and measured at 320 nm at different time points over a period of 3 h.

### 2.5. Insulin Loading Efficiency of Ba Salt Particles

#### 2.5.1. Insulin Loading Efficiency of Ba Salt Particles and Subsequent Release of Insulin from Particles at a Wide Range of pHs

FITC-insulin-loaded Ba salt particles were prepared as stated in Section 2.2.2. After 30 min of incubation at 37 °C, 200 µL of DMEM prepared at different pHs (pH 7.78 to pH 1.78) was added. After incubation at room temperature for 10 min, the samples were centrifuged at 6000 rpm for 2 min at 4 °C. The supernatant was discarded and the resultant pellet was washed with 50 µL of HEPES, 2X. At the end of washing the supernatant was again discarded and the pellet carrying insulin-bound particles was re-suspended with 200 µL of 10 mM EDTA.

200 μL of supernatant was carefully transferred into a black 96-well plate (PerkinElmer Opti-Plate™-96 F, Waltham, MA, USA) and fluorescence intensity values were recorded using a fluorescence microplate reader (PerkinElmer Victor X5 2030 Multi-label Reader, (Waltham, MA, USA) set with excitation/emission filters at 485 nm/535 nm wavelengths.

A standard curve was prepared from absorbance values for known amounts of free FITC-Insulin (0, 400, 800, 1200, 1600, 2000, 4000 ηg) added to DMEM. % loading at pH 7.78 was calculated from the formula given below.
(1)$$\%\;\mathrm{loading}=\frac{\mathrm{FITC}-\mathrm{Insulin}\;\mathrm{conc}.\mathrm{in}\;\mathrm{pellet}}{\mathrm{conc}.\;\mathrm{of}\;\mathrm{added}\;\mathrm{FITC}-\mathrm{Insulin}}\times 100$$
% release at subsequent pHs (7.18–1.78) was calculated by subtracting the amount of FITC-Insulin found in the pellet from the amount of FITC-Insulin added initially and finally, multiplying by 100.
(2)$$\begin{array}{l} \%\mathrm{release}\;\mathrm{of}\;\mathrm{Insulin}\;\\ \frac{=(\mathrm{FITC}\;-\;\mathrm{Insulin}\;\mathrm{conc}.\mathrm{in}\;\mathrm{pellet}\;\mathrm{synthesized}\;\mathrm{at}\;\mathrm{pH}\;7.78\;-\;\mathrm{FITC}\;-\;\mathrm{Insulin}\;\mathrm{conc}.\mathrm{in}\;\mathrm{pellet}\;\mathrm{of}\;\mathrm{particles}\;\mathrm{synthesized}\;\mathrm{at}\;\mathrm{subsequent}\;\mathrm{pH})\;\times 100}{\mathrm{FITC}\;-\;\mathrm{Insulin}\;\;\mathrm{conc}.\mathrm{in}\;\mathrm{pellet}\;\mathrm{of}\;\mathrm{NP}\;\mathrm{synthesized}\;\mathrm{at}\;\mathrm{pH}\;7.78} \end{array}$$

#### 2.5.2. Morphological Assessment of Empty and Insulin-Loaded Particles by FE-SEM

The Ba salts particles were similarly prepared as described in the earlier Section 2.3.2 for FE-SEM and EDX analysis (Section 2.3.2). Insulin-loaded particles were prepared separately (as stated in Section 2.2.2). Visualization and documentation were done using FE-SEM (Hitachi/SU8010, Tokyo, Japan) at 2.0 kv.

### 2.6. Assessment of Adhesion of Ba Salt Particles to Mucin

#### 2.6.1. Spectrophotometric Analysis of Particle Adhesion to Mucin

Ba salt particles were prepared as stated in the particle preparation Section 2.2.1. 200 µL of milliQH_2_O was added followed by the addition of 100 µL of mucin (3 g/L). The mixture was then incubated at 37 °C for 10 min followed by centrifugation at RPM13, 200 for 5 min. The supernatant was collected in a fresh tube. 250 µL of Bradford agent was directly added to the tube. The mixture was then incubated for 5 min before transferring it to the 96 well microplates. Absorbance was read at 595 nm using victor X5 spectrophotometer (PerkinElmer, Waltham, MA, USA). A standard curve was created using a series of different concentrations of mucin solution of 25 to 600 µg, mixed with 250 µL of Bradford agent. An unknown amount of mucin present in supernatant was calculated from the standard curve. % adhesion of particle to mucin was then calculated from added mucin with the two-step formula below:(3)$$\begin{array}{l} \mathrm{Mucin}\;\mathrm{bound}\;\mathrm{to}\;\mathrm{Particles}=\mathrm{mucin}\;\mathrm{present}\;\mathrm{in}\;\mathrm{the}\;\mathrm{pallet}\\ \;\;\;\;\;\;=\mathrm{added}\;\mathrm{mucin}-\mathrm{mucin}\;\mathrm{from}\;\mathrm{supernatant} \end{array}$$
(4)$$\%\mathrm{adhesion}=\frac{\mathrm{mucin}\;\mathrm{bound}\;\mathrm{to}\;\mathrm{particles}\times 100}{\mathrm{added}\;\mathrm{mucin}\;}$$

#### 2.6.2. Assessment of Particle Adhesion to Mucin Using FT-IR

Samples were prepared in bigger volumes for this experiment. 250 µL of 1M of BaCl_2_ in a 50 mL tube was mixed with 100 µL of 1M Na_2_SO_4_ or Na_2_SO_3_ or Na_2_CO_3_ to prepare BaSO_4_, BaSO_3_, or Ba_2_CO_3_ particle. The samples were then incubated at 37 °C for 30 min. Sample tubes were then topped up to 25 mL with the addition of milliQH_2_O, followed by the addition of a solution containing 5 mL of mucin (3g/L). Samples were then centrifuged for 15min at 5000 RPM. Precipitates were then stored at −20 °C overnight, followed by storing at −80 °C for 0.5 h and freeze-drying (Labconco freeze dryer, Kansas City, MO, USA) for 5 h. Varian FT-IR (Santa Clara, CA, USA) and Varian Resolution Pro 640 software (Santa Clara, CA, USA) were used to check the spectra of particles alone, particle-mucin complexes, and mucin alone.

### 2.7. Administration of Insulin-Loaded Particles to Diabetic Animals and Management of Hyperglycemia

#### 2.7.1. Induction of Diabetes in Rats with Streptozotocin (STZ)

Animal experiments were conducted in accordance with the approved Animal ethics by Monash Animal Ethics Committee (MUM/2018/13). 8–12 weeks-old male healthy Wister Kyoto rats (WKY) were subjected to intraperitoneal (IP) injection of STZ (65 mg/kg). A fraction of the rats developed hyperglycemia with a persistent high level of peripheral blood glucose (≥13 mM) within a week or two. Glucose levels in the blood (collected via tail vein) were measured using a glucometer (Terumo, Hatagaya, Shibuya-ku, Japan). The rats that didn’t develop diabetes with 1st injection were given a 2nd IP dose of 65 mg/kg STZ.

#### 2.7.2. Effect of Orally Administrated Insulin-Loaded Particles on Hyperglycemia

Rats showing a clear sign of diabetes (peripheral blood glucose level of ≥13 mM) were divided into different control and treatment groups. Each group consists of three animals. After taking a baseline blood glucose reading, each rat was given Insulin aspart-loaded Ba salt formulations via oral gavage. Rats from the control group were kept untreated (negative control), whereas rats from the positive control group received a solution containing free insulin (100 IU/kg). Treatment groups received 500 µL of the solution containing insulin-loaded particles (BaSO_4_, BaSO_3_, or BaCO_3_ particles) by gavage. Blood glucose was read at regular time intervals (0.5 h, 1 h, 2 h, 3 h, and 4 h).

Blood glucose level at “0” h is the baseline blood glucose reading taken right before oral delivery of insulin-loaded particles. % reduction at any given point of time was calculated from baseline blood glucose level using the below formula,
(5)$$\begin{array}{l} \mathrm{reduction}\;\mathrm{of}\;\mathrm{blood}\;\mathrm{glucose}\;\mathrm{level}\\ =\frac{\left(\mathrm{baseline}\;\mathrm{blood}\;\mathrm{glucose}\;\mathrm{read}\;-\;\mathrm{blood}\;\mathrm{glucose}\;\mathrm{reading}\;\mathrm{at}\;\mathrm{any}\;\mathrm{point}\;\mathrm{of}\;\mathrm{time}\right)\times 100}{\mathrm{baseline}\;\mathrm{blood}\;\mathrm{glucose}\;\mathrm{level}} \end{array}$$

#### 2.7.3. Statistical Analysis

The *p*-value was calculated by applying one-way ANOVA to confirm any significant reduction of blood glucose level compared to pre-treatment baseline blood glucose level, at any point in time after treatment.

## 3. Results

### 3.1. Chemical Characterization and Morphological Analysis of Synthesized Ba Salt Particles

#### 3.1.1. Fourier Transform Infrared Spectroscopy (FT-IR)

FT-IR was used to confirm the formation of Ba salt particles (BaSO_4_/BaSO_3_/BaCO_3_) by revealing the functional groups. Figure 3 displays the FT-IR spectrum obtained for three different Ba salt particles. IR spectrum for BaSO_4_ (Figure 3a) shows peaks at 1188 cm^−1^, 1060 cm^−1,^ and 982 cm^−1^ which are attributed to symmetric stretching of SO_4_^2−^, peaks at 1635 cm^−1^ due to stretching vibration of SO_4_^2−^ and peaks at 604 cm^−1^ and 637 cm^−1^, representing out-of-plane bending for SO_4_^2−^. Peaks at 2360 cm^−1^ seem to be the overtone of S-O vibration [24,25]. The existence of the functional group suggests the generation of BaSO_4_ particles. Figure 3b shows the characteristic peaks for SO_3_^2−^ found at 490 cm^−1^, 630 cm^−1^, 912 cm^−1^, and 1133 cm^−1^ position, suggesting the formation of BaSO_3_ particles [26,27]. The characteristic peaks for CO_3_^2−^ found at 855 cm^−1^ and 692 cm^−1^ (Figure 3c) were owing to in-plane and out-of-plane bending, whereas the peak at 1414 cm^−1^ corresponds with asymmetric stretching of C–O bond and peak at 1059 cm^−1^ is attributed to the symmetric C–O stretching vibration [28], thus indicating the formation of BaCO_3_ particles.

#### 3.1.2. Element Analysis of Ba Salt Particles by Energy Dispersive X-ray Spectroscopy (EDX)

EDX was carried out for elemental analysis of Ba salt particles (Figure 4). The presence of the desired elements in the samples would confirm the formation of the respective particles. Elemental analysis of BaSO_4_ (Figure 4a) and BaSO_3_ (Figure 4b) shows the presence of Ba, S, and O, whereas analysis of BaCO_3_ confirms the presence of Ba, C, and O (Figure 4c).

### 3.2. Particle Stability Assessment

One of the biggest challenges in oral delivery is pre-systemic degradation due to harsh pH in the stomach having a fluctuating pH which can be extremely acidic (~pH 1) in fasting state. Therefore, any oral delivery agent needs to be resistant to extreme acidic pH to survive during GIT transport. To test how stable Ba salt particles are in different pHs, a set of in vitro experiments were designed and carried out by exposing the formulations to a wide range of pHs for a period of 3 h.Figure 5 shows changes in the turbidity pattern at 320 nm for Ba salt particles. Both BaSO_4_ and BaSO_3_ particles demonstrated much higher absorbance than BaCO_3_ particles when measured immediately after their generation at pH 7.78, reflecting the tendency of BaSO_4_ and BaSO_3_ to generate a significantly higher number of particles than BaCO_3_. Consistently high turbidity values for BaSO_4_ particles indicate their excellent resistance to degradation over a wide range of pHs at different time points (1–3 h), while the turbidity of BaSO_3_ particles sharply dropped at pH 1.69 suggesting the considerable amount of particle loss. BaCO_3_ maintained almost the same level of turbidity throughout the range of pHs, with apparently no significant particle loss at low pH.

### 3.3. Insulin Loading of Ba Salt Particles

#### 3.3.1. Insulin Loading Efficiency of Ba Salt Particles and Subsequent Release of Insulin from Particles at a Wide Range of pHs

The affinity of insulin molecules towards different Ba salt particles was assessed by separating FITC-conjugated insulin-loaded particles from free fluorescent insulin by centrifugation. Fluorescence intensity was measured at 485 nm/535 nm (excitation/emission) wavelengths. The amount of FITC-insulin present in the pellet was calculated from the standard curve prepared with a known amount of FITC-insulin vs. respective absorbance. Appendix A shows the standard curve prepared with free FITC-insulin. % loading was calculated by dividing the amount of FITC-insulin present in the pellet with the amount initially added and then multiplying with 100. Figure 6a shows % of insulin binding to three different Ba salt particles, with BaCO_3_ particles demonstrating a higher binding affinity for insulin (100%) than BaSO_4_ and BaSO_3_ particles. The equations used to calculate the % of loading and % of release of insulin were shown in Figure 6b. FITC-insulin-loaded particles were exposed to a wide range of pHs, and as shown in Figure 6c, nearly 20%, 50%, and 60% of insulin were released from BaSO_4_, BaCO_3_, and BaSO_3_ particles, respectively, when the respective insulin-particle complexes were exposed to pH of 2.47, whereas at a pH of 1.78, only around 30% of insulin was released from BaSO_4_ and approximately 80% of insulin was released from BaSO_3_ particles.

#### 3.3.2. Morphological Analysis of Ba Salt Particles by FE-SEM

FE-SEM analysis was carried out to study the particle morphology with respect to particle size, shape, and aggregation pattern with and without insulin loading into the particles. Figure 7 shows micrographs for 3 different types of Ba salt particles. (a) BaSO_4_ particles possessed distinctive irregular, oval to hollow rounder shape, with the bigger particles having oval to hollow rounder shape with particle size between 110–200 nm and the smaller ones being irregularly shaped with particle size between 30–100 nm. (b) BaSO_3_ particles also had the prominent distribution of big (90–270 nm) and very small (15–40 nm) sized particles. The particles were found to be rounder shaped and held in tight baseball-shaped clusters with a rough surface area. Each distinctive particle cluster ranged from 500 nm to 1.0 µm size in diameter. Bigger particles displayed rounder shape morphology with smooth surface area, without forming a cluster (c) BaCO_3_ particles demonstrated distinctive square to rectangular shaped morphology, with the particle size varying widely from 50 to 500 nm. Particles were closely aggregated with one another, forming rod/filament-like structures.

### 3.4. Analysis of Effect of Insulin Loading into Particles by FE-SEM

Figure 8, Figure 9 and Figure 10 show the scanning electron microscopy images of empty Ba salt particles and insulin-loaded particles placed side-by-side to observe any morphological changes in the context of size, shape, and aggregation patterns as a result of insulin loading.

#### 3.4.1. Effect of Insulin Loading into Baso_4_ Particles: Changes in Morphology and Aggregation Pattern

Insulin was loaded into BaSO_4_ particles in varying amounts. Loading of insulin altered the shape of the particles, with different concentrations of insulin showing a different degree of structural changes in particle morphology (Figure 8).

Figure 8a shows that free BaSO_4_ particles are of rounder shape with a distinctive hollow structure in the middle and a smooth outer surface. Figure 8b shows BaSO_4_ particles prepared with 2 µg of initially added insulin, giving the impression of free particles residing with insulin-loaded particles which were morphologically different from the empty particles. Insulin-loaded particles are baseball-shaped with a rough surface area and without a hollow structure. Figure 8c,d show insulin-loaded BaSO_4_ particles prepared with 10 µg and 50 µg of insulin. With increasing insulin concentrations, the particle looked more elongated-shaped and the surface area appeared rougher. Interestingly, there was no empty BaSO_4_ particle visualized in a high concentration of insulin.

Different amounts of insulin loading show different degrees of structural changes in particle morphology. (**a**) Free particles were of rounder shape with a distinctive hollow structure in the middle and a smoother outer surface. (**b**) A low amount of insulin (2 µg) added prior to particle formation showed the presence of characteristic free particles along with few insulin-loaded particles with a distinctive change in morphology. (**c**) and (**d**) Insulin-loaded particles prepared with 10 µg and 50 µg of initially added insulin. With an increasing amount of insulin, particles became more oval-shaped and the surface area looked rougher. Pictures were taken at 2.0 kV.

#### 3.4.2. Effect of Insulin Loading into Baso_3_ Particles: Change in Morphology and Aggregation Pattern

Figure 9 shows morphological features of empty and insulin (FITC-insulin)-loaded BaSO_3_ particles. There were no noticeable changes in size, shape, or surface morphology of a single particle. However, a very distinctive change in aggregation pattern was observed. Empty particles tended to get clumped together, whereas insulin-loaded particles looked much less aggregated, with their clusters clearly oval shaped.

#### 3.4.3. Effect of Insulin Loading into Baco_3_ Particles: Changes in Morphology and Aggregation Pattern

Figure 10 shows morphological features of empty and insulin-loaded BaCO_3_ particles with no noticeable changes in size, shape, and surface morphology of individual particles. However, very distinctive patterns in the aggregation pattern could be observed. Free particles tended to get clumped together, whereas insulin-loaded particles looked much less aggregated.

### 3.5. Assessment of Mucin Adhesion to Ba Salt Particle

In vitro tests were done to determine the affinity of different Ba salt particles for mucin to predict whether the particles would be capable of crossing the intestinal lining. The data from the FT-IR analysis would indicate any possible interactions of mucin molecules with the particles. Mucin adhesion to the particles was further assessed by the spectrophotometric protein quantification method.

#### 3.5.1. Analysis of Mucin Adhesion to Ba Salt Particles Using FT-IR

FT-IR was used to confirm the mucin binding of Ba salt particles (BaSO_4_/BaSO_3_/BaCO_3_) by revealing the functional groups for particle along with mucin protein. Samples for particles only, mucin only, and particles adhered to mucin were analyzed with FT-IR. The comparison of the peak patterns showed positive mucin binding for all three Ba salt particles. Free mucin displayed characteristic peaks of N-H at 3279 cm^−1^, Amide I at 1634 cm^−1^, Amide II at 1549 cm^−1^, C-H at 1434 cm^−1^, and 1374 cm^−1^, Amide III 1232 cm^−1^, C–O–C at 1115 cm^−1^ and C–C–O at 1029 cm^−1^ [29]. Figure 11a reveals FT-IR spectrum of BaSO_4_ particles alone, free mucin, and BaSO_4_ particle-mucin complexes. BaSO_4_ particles showed 8 characteristic peaks as described in an earlier Section 3.1.1. On the other hand, mucin displayed 8 characteristic peaks (as found in the free mucin sample). Mucin-adhered BaSO_4_ particles demonstrated 12 distinctive peaks, with peaks 2360 cm^−1^, 1188 cm^−1^, 1064 cm^−1^, 982 cm^−1^, 635 cm^−1^, and 605 cm^−1^ matching the peaks of BaSO_4_ particles in the similar positions. The peak at 3415 cm^−1^ showed a deviation to 2392 cm^−1^ probably due to mucin attachment to the particles. Similarly, mucin peaks at 3279 cm^−1^, 1549 cm^−1^, and 1434 cm^−1^ seemed to be shifted to 3275 cm^−1^, 1539 cm^−1^, and 1439 cm^−1^ respectively in mucin-attached particles. The existence of characteristic peaks for BaSO_4_ and mucin indicates interactions between BaSO_4_ particles to mucin.

Figure 11b shows the FI-IR spectrum for free BaSO_3_ particles, free mucin, and BaSO_3_-mucin complex. BaSO_3_ particles possess 4 characteristic peaks as described in the previous Section 3.1.1. On the other hand, mucin has 8 characteristic peaks (as shown in the free mucin sample). Mucin-adhered BaSO_3_ particles showed 13 distinctive peaks. The peaks of BaSO_3_ particles, 1133 cm^−1^, 912 cm^−1^, 630 cm^−1^ and 409 cm^−1^ were found at 1131 cm^−1^, 919 cm^−1^, 509 cm^−1^, and 474 cm^−1^ positions in mucin-adhered BaSO_3_ particles. On the other hand, 8 characteristic peaks from mucin were found at 3311 cm^−1^, 1648 cm^−1^, 1539 cm^−1^, 1434 cm^−1^, 1298 cm^−1^, 1343 cm^−1^, 1187 cm^−1^, and 1101 cm^−1^ positions. Mucin peaks showed deviation from their positions apparently due to its binding to BaSO_3_ particles. The presence of characteristic peaks for BaSO_3_ particle as well as for mucin indicates positive adhesion of BaSO_3_ particles to mucin.

Figure 11c shows the spectrum for free BaCO_3_ particles, free mucin, and BaCO_3_-mucin complex. BaCO_3_ particles showed 4 characteristic peaks as described in the earlier Section 3.1.1. Mucin-adhered BaCO_3_ particles demonstrated 9 distinctive peaks. Mucin characteristic peaks at 3279 cm^−1^, 1634 cm^−1^, 1549 cm^−1^, 1232 cm^−1^, and 1115 cm^−1^ were found, respectively, at 3384 cm^−1^, 1645 cm^−1^, 1555 cm^−1^, 1213 cm^−1^, and 1121 cm^−1^ positions in mucin-adhered BaCO_3_ particles. The presence of characteristic peaks for BaCO_3_ particles and mucin indicates positive adhesion of BaCO_3_ particles to mucin.

#### 3.5.2. Quantitation of Mucin Adhesion to Particles

Percentage (%) of mucin adhesion to different Ba salt particles was calculated by measuring the protein content (mucin) precipitated out with the particles after centrifugation. Protein quantification was done by the Bradford method. The standard curve was prepared with a series of a known amount of mucin versus respective absorbance values. Figure 12 presents Bradford assay data for mucin adhesion, where (a) shows the standard curve prepared from abs vs. known amount of mucin, whereas (b) reveals the % of adhesion to 3 different types of Ba salt particles. BaSO_4_ particles showed highest mucin adhesion (100%), followed by BaSO_3_ and BaCO_3_ particles with 60–70% of mucin adhesion.

### 3.6. Oral Administration of Insulin-Loaded Particles to Diabetic Rats and Effect on Blood Glucose Level

Short-acting human insulin analog, Insulin Aspart (NovoRapid, Novonordisk)-loaded Ba salt particles were administrated to STZ-induced diabetic male Wister Kyoto (WKY) rats. For oral administration, particles prepared with a high concentration of reactant salts (20× higher than that used in original formulations) were loaded with a high dose of insulin aspart (100 IU/kg).

#### Effect of Orally Administrated Insulin on Hyperglycemia

Insulin aspart loaded into Ba salt particles were administrated to STZ-induced diabetic male WKY rats. For oral administration, Ba salt particles prepared with 20X concentrated reactant salts (Table 2) were loaded with a high dose of insulin aspart (NovoRapid^®^, Novo Nordisk, Bagsværd, Denmark) (100 IU/kg). Figure 13 shows the data from control groups and treatment groups. Control groups consist of no treatment (negative control) and oral treatment of free insulin (100 IU/kg) (positive control). As expected, none of the groups showed any significant reduction of blood glucose level (*p* < 0.05) at any time point in the study. For BaSO_4_ particles with loaded insulin, blood glucose level was significantly low compared to the baseline level in 1–3 h (*p* < 0.05). At 4th h, the level was still low compared to the baseline level but not significant, with an ascending trend. BaCO_3_ particles with loaded insulin showed a significant decrease of blood glucose level at 1–2 h, although blood glucose level started to show a slight rise at 3rd h, and by 4th h, it was back to the baseline level. However, insulin-loaded BaSO_3_ particles showed a reduction in blood glucose level but the reduction was not significant at 4 h.

Figure 14 shows a calculated % reduction of blood glucose level for BaSO_4_, BaSO_3_, and BaCO_3_ loaded insulin. % reduction calculation shows a significant reduction of blood glucose level for BaSO_4_ loaded insulin from 1–3 h and BaCO_3_ loaded insulin between 1–2 h (*p* < 0.05). Again, because of inconsistency in glucose reduction pattern in different animals, % of reduction in case of BaSO_3_ loaded insulin treatment was not significant at any point in time.

## 4. Discussion

Oral insulin is the most attractive choice to eliminate the necessity of subcutaneous insulin therapy [30]. As mentioned earlier, the most promising concept for oral insulin delivery is based on attaching the insulin molecule to a “cargo” that carries it through the GI tract. For that, the “cargo” or “carrier” itself has to pass all the criteria for an oral delivery candidate. Several papers reported polymeric nanoparticle as carriers for insulin [3,4,5,6]. The concept of complexing insulin molecules with inorganic salt particles based on ionic interactions is a relatively new one, having enormous potential [22]. Ba salt particles (BaSO_4_, BaSO_3,_ and BaCO_3_) have both positive and negative charged domains conferred, respectively, by cations (Ba^2+^) and anions (SO_4_^2−^, SO_3_^2−^, and CO_3_^2−^). Due to heterogeneous charge distribution, these particles are capable of binding peptide molecules like insulin through electrostatic interactions. The target protein molecule here is insulin (5.8 kDa) which also has localized surface charges either positive or negative at basic or acidic pH. These surface charges of insulin likely allowed it to electrostatically bind with either cationic or anionic charges of Ba salt particles, depending on the pH of the solution. However, the challenge was whether Ba salt particles could resist a harsh GIT environment and allow insulin molecules to pass through the intestinal wall [31,32]. This is the first study to explore the potential of Ba salt particles as oral delivery carriers for insulin.

A series of in vitro tests were carried out in GIT-mimicking conditions, before investigating the effect of orally administered insulin-loaded particles on hyperglycemia in diabetic Wister–Kyoto rats (WKY). GIT itself is a large and complex organ system where each different chamber has its own pH and enzymatic environment. Orally administrated protein therapeutics often show poor bioavailability due to the physical as well as physiological barriers imposed by the GIT. One of the biggest challenges is protein degradation by the extreme acidic pH of the stomach. The second prominent barrier is the enzymatic degradation in both the stomach and intestine. Even after successfully bypassing the stomach pH and enzymatic action, the protein-loaded particles or the released proteins are supposed to encounter mucin of the GI lining [33], prior to crossing the epithelium either via transcellular or paracellular route to reach the blood circulation.

### 4.1. Characterization and Morphological Screening of Empty Ba Salt Particles

Three different Ba salt particles were characterized with FTIR to identify the functional groups of the respective particles, and EDX to detect the various elements present in those particles, thus confirming the identity of each particle type.

### 4.2. Particle Stability Assessment

Particle stability study was important to understand how efficiently Ba salt particles could resist harsh pHs, and thus, predict which of the Ba salt particles among BaSO_4_, BaSO_3_, and BaCO_3_ would show the highest potential as an oral carrier. Successful oral delivery can only take place if the formulations are resistant enough in fluctuating stomach pH. Another important factor in oral delivery is stomach and intestinal residence time. Depending on the particle properties, the residence time inside the gut could vary from 20 min to 3 h [34]. An ideal oral delivery carrier must be resistant to both basic and acidic pHs within that time-frame. The stomach has fluctuating pH which can vary from pH 1.7 to 4.7 [35], whereas intestinal pH is around 6–8 [35].

In vitro tests were done to see whether the particles are capable of sustaining in a wide range of pHs over a certain period. As mentioned at the beginning, these particles are new and at the time of the study, properties were mostly unknown. It was important to check their response towards pH and enzymatic activity separately as well as combined. DMEM media composition is suitable for sustaining any biological molecules, like, proteins. At the same time, DMEM can accommodate pH change to a wide range. The particles (free and insulin bound) were exposed to DMEM prepared with a wide range of pH to study their self-resistance as well as resistance to hold a protein molecule against drastic pH change (described in Section 4.3). Turbidity which is measured as absorbance at 320nm increases as particle formation is accelerated and decreases as particle formation is inhibited or particle dissolution takes place. Particles were prepared at pH 7.8 and then exposed to lower pHs. The in vitro tests showed that BaSO_4_ particles had the highest synthesis rate at pH of 7.8, as well as the best resistance at all pHs throughout the period of 3 h, followed by BaCO_3_ and BaSO_3_ particles. All three formulations seemed to be suitable as oral delivery carriers based on their pH resistance.

### 4.3. Insulin Loading Efficiency of Ba Salt Particles and Subsequent Release of Insulin from Particles at Lower pHs

The fluorometric assay was performed with a fixed amount of FITC-insulin added prior to the formation of insulin-loaded particles. The portion of the added FITC-insulin absorbed by the particles were later separated in a pellet form by centrifugation. All 3 Ba salt particles showed very good insulin loading efficiency (80–100%). And insulin-loaded particles were found to be stable while gradually reducing pH to 5.0, with almost no release of insulin from the complex. At pH < 5 different degrees of insulin release were observed. Insulin release from BaSO_4_ particles was 30% even in very harsh acidic pH, implying that this particular nano-insulin formulation would be stable inside the stomach regardless of fed or fasting state. However, BaSO_3_ and BaCO_3_ particles with loaded insulin showed almost 80–100% release in acidic pH of nearly 1.0.

The capability of the Ba salt particles in protecting protein molecules was further assessed using albumin protein against different pHs and enzymatic effects. The data generated with albumin has been presented in supplementary figures (Appendix A) and discussed in detail.

### 4.4. Morphological Analysis of Ba Salt Particle by FE-SEM

FE-SEM image analysis confirmed the formation of Ba salt particles and revealed their morphology with regards to size, shape, and aggregation pattern. The images also showed the morphological changes of particle clusters while being loaded with insulin, confirming successful insulin loading of the particles. Appendix A are added with higher resolution micrographs, demonstrating in detail the morphological changes of BaSO_4_ particles owing to insulin binding.

### 4.5. Assessment of Mucin Adhesion to Ba Salt Particles

Mucin of GI lining is considered as the primary barrier for any molecule to cross the intestinal lining and reach the systemic circulation. Indeed, secreted mucin along with the contents of the gut creates an intestinal fluid environment [36]. Therefore, orally delivered insulin-loaded Ba salt particles will be in contact with mucin while crossing the intestinal lining. In vitro tests were done to predict whether the particles would be capable of crossing the intestinal lining. Particle adhesion to mucin was analyzed with FT-IR and Bradford protein assay kit, with FT-IR bands showing characteristic peaks for both Ba salts particle and mucin, and the protein assay reconfirming mucin adhesion to the particles quantitatively. All three Ba salt particles were found to have high mucin adhesion (60–100%).

### 4.6. Effect of Orally Administrated Nano-Insulin on Hyperglycemia

For oral delivery, particle formulations were prepared in 20× concentrated form to efficiently load high insulin (100 IU/kg) dose. All three formulations of Ba salt particles with loaded insulin showed a significant reduction in blood glucose levels. The effect generated during oral administration of insulin-loaded particles was surprisingly similar to the effect noticed for subcutaneous delivery of commercial human insulin aspart, only lasting 4–5 h with a maximum drop in blood glucose level during 2–3 h. The onset of the action on hyperglycemia started at 1 h for the oral nano-insulin formulation, which could be due to the required transport time for insulin-loaded particles to reach the blood stream. The percentage of reduction of blood glucose level in comparison to the baseline level showed the maximum reduction below 50% at any time point for all 3 Ba salt particles. The newly designed Ba salt/Insulin showed a visible significant effect on hyperglycemia at doses of 100 IU/Kg and % reduction in blood glucose is less than 50%.

In vitro insulin loading and mucin adhesion profile were promising from all three Ba salt particles, however, in vitro stability and protection from pH and enzymatic activity data showed stronger protection from BaSO_4_ and BaCO_3_ particles towards protein molecule compared to BaSO_3_ particle. In vivo activity in diabetic rats also reflected that. Although the BaSO_3_ particle also successfully reduced blood glucose levels, there were fluctuations and inconsistencies in the reduction pattern between the animals which ultimately affected the significance of the finding. Notably, this inconsistency was also observed for the other two particles in a lower dose of insulin (50 IU/kg, data not shown here). The authors believe BaSO_3_ loaded insulin might show a significant and consistent reduction of blood glucose at much higher doses. In summary, BaSO_3_ loaded insulin is less efficient compared to BaSO_4_ and BaCO_3_ particle loaded oral insulin. However, there is plenty of room for improvement for these formulations.

A common problem with potential oral insulin formulations from the literature is the low bioavailability and requirement for very high doses [6,7,37]. This is because of degradation in the GI tract and low percentage of intestinal crossing mainly. The onset, duration of action, and maximum reduction of blood glucose level vary a lot from formulation to formulation due to properties and diversified mechanism of action of the particle or system used as DDS for insulin molecule. Wu et.al. reported, insulin loaded PLGA/RS nanoparticles (50 IU/kg) showed a slow decrease after a lag phase of 4 h with no visible effect. The maximum blood glucose reduction was in 10 h (32%) and the effect lasted for 24 h [6]. Pan et al. (2002) reported the effectiveness of positively charged CS-NP that was bound to insulin at doses 21 IU/kg and hypoglycemia prolonged for 15 h. There was no reporting of % decrease of blood glucose level [7]. Another nanoparticle (NP) loaded insulin system composed of chitosan (CS) and poly(g-glutamic acid) reported an extended period of activity of insulin aspart when loaded in NP. NP loaded insulin aspart was given orally in a similar manner to Ba salt insulin administration reported in this paper. However, the effect was observed as peak-less prolonged reduction of blood glucose. The formulation loaded with insulin aspart, which is originally a short-acting human insulin analog. Interestingly in NP loaded form oral administration of it showed an effect similar to subcutaneous intermediate-acting formulations available in the market. The effect was an observer at doses of 30 IU/kg and reduction of blood glucose level was never more than 30–40% [37].

There could be several factors involved that can influence any nanoparticle-mediated insulin formulation. In general nanoparticle binding towards protein is effected/influenced by surface charge, crystallinity, size of nanoparticle as well as insulin molecule. On the other hand, oral delivery of nanoparticle loaded insulin depends on size, lipophilicity, surface charge, and resistance of the carrier molecule. The reported Ba salt insulin formulations are at an early stage. As new formulations, they are in vivo profile of GI transport and activity in the blood are unknown. Although, quick onset with the same duration of action as native insulin aspart indicates a unique in vivo profile. The formulations are under study for their details on in vivo mechanisms of action and improvement of pharmacokinetic parameters at present.

## 5. Conclusions

As a novel group of oral macromolecular drug delivery system, Ba salt particles, in particular, BaSO_4_ and BaCO_3_ look very promising. In vitro tests on binding, resistance and characterization have shown that the particles can bind protein therapeutics like insulin efficiently. In vivo study on rats has proven their effectiveness in managing hyperglycemia when given orally. Novel nano-insulin, i.e., Ba salt particle-loaded insulin aspart (oral) has a similar effect on blood glucose level as the original insulin aspart subcutaneous formulation in higher doses. Therefore, the developed oral formulations could be an attractive alternative to subcutaneous ones in the future.

## Figures and Tables

**Figure 1 pharmaceutics-12-00710-f001:**
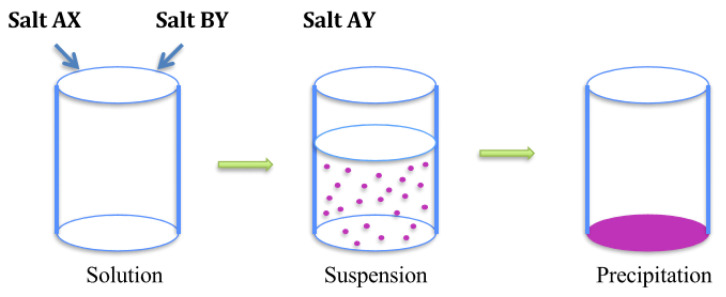
A simplified schematic representation of precipitation reaction [24]. AX(soluble) + BY(soluble) = AY(precipitate) + BX(soluble)

**Figure 2 pharmaceutics-12-00710-f002:**
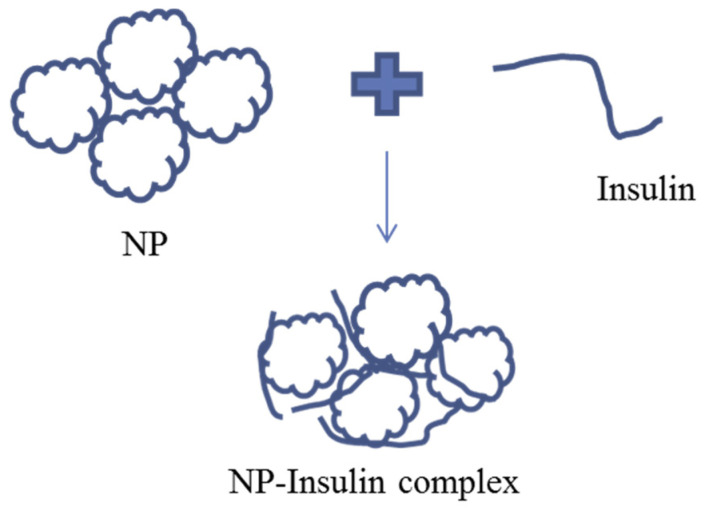
A simplified illustration of insulin loading into particles.

**Figure 3 pharmaceutics-12-00710-f003:**
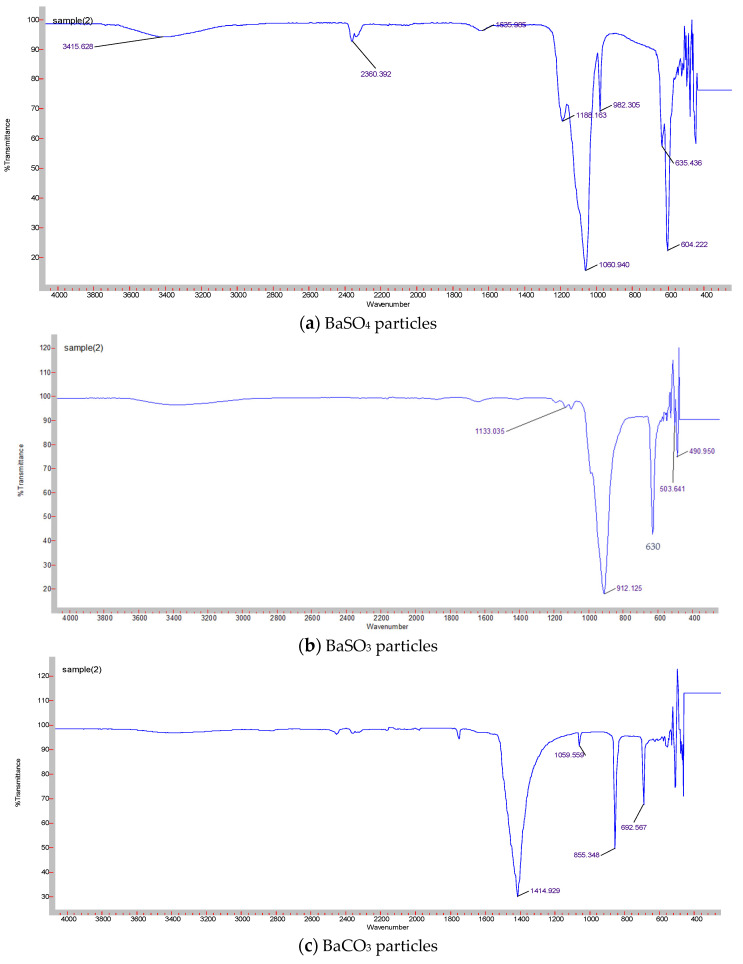
FT-IR analysis of Ba salt particles. (**a**) Shows characteristic peaks for BaSO_4_ particles. Characteristic peaks are detected at 2360 cm^−1^, 1188 cm^−1^, 1060 cm^−1^, 982 cm^−1^, 1635 cm^−1^, 604 cm^−1^, and 637 cm^−1^. (**b**) Represents characteristic peaks for BaSO_3_ particles at 490 cm^−1^, 630 cm^−1^, 912 cm^−1^, and 1133 cm^−1^. (**c**) Represents characteristic peaks for BaCO_3_ particles at 855 cm^−1^, 692 cm^−1^, 1414 cm^−1^, and 1059 cm^−1^.

**Figure 4 pharmaceutics-12-00710-f004:**
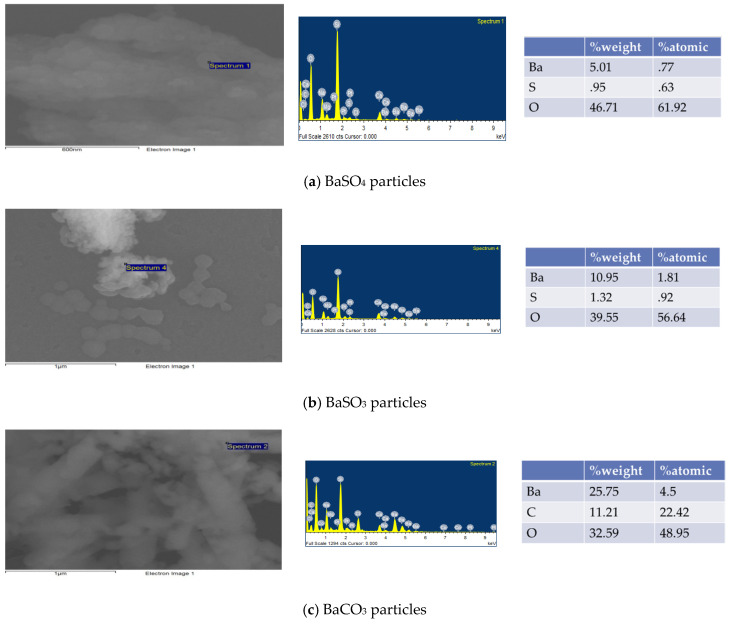
Element analysis of Ba salt particles. (**a**) The presence of Ba, S, and O elements in the sample for BaSO_4_ particles. (**b**) Presence of Ba, S, and O elements in the sample for BaSO_3_ particles. (**c**) The presence of Ba, C, and O elements in the sample for BaCO_3_ particles.

**Figure 5 pharmaceutics-12-00710-f005:**
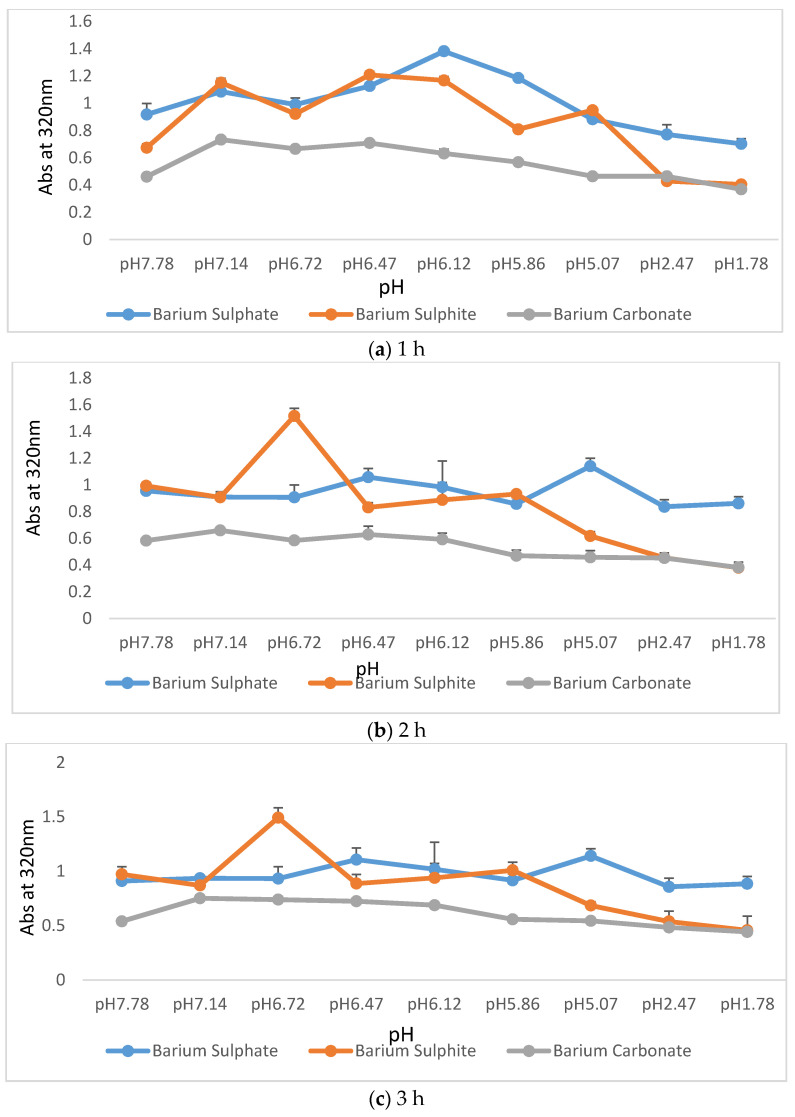
Dissolution of profiles of three different Ba salt particles upon exposure to a wide range of pHs (pH 7.78–pH 1.78) over a time period of 3 h. Ba salt particles were synthesized at pH 7.78 and exposed to different pHs to observe particle dissolution. (**a**), (**b**) and (**c**) plots data from 1 h, 2 h and 3 h respectively.

**Figure 6 pharmaceutics-12-00710-f006:**
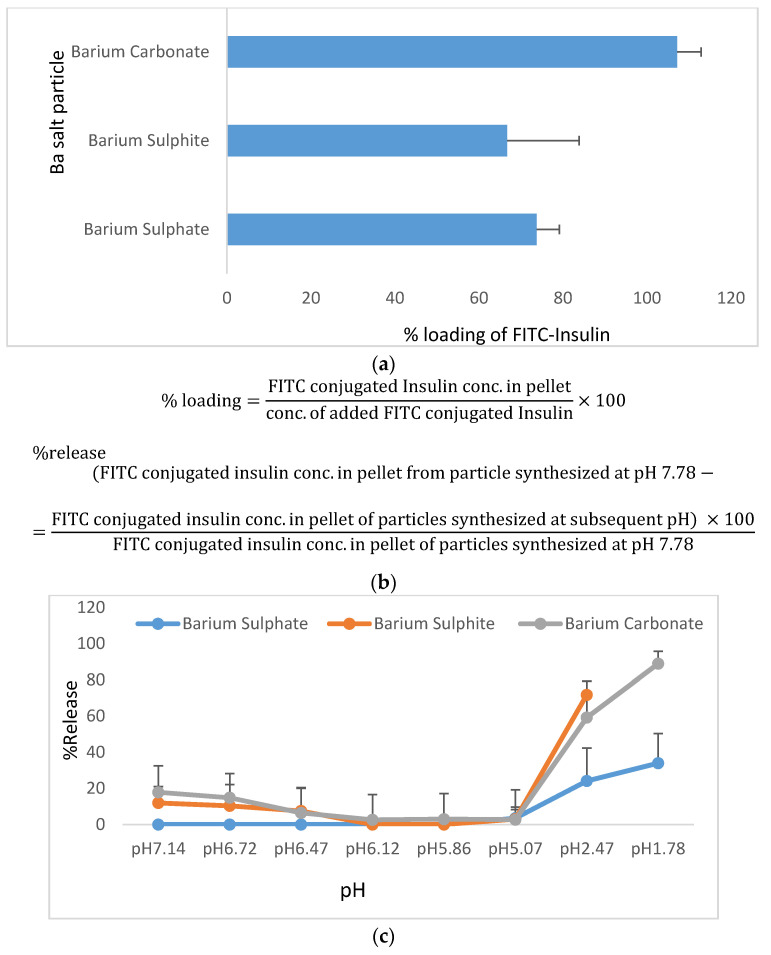
Measurement of insulin binding affinity towards Ba salt particles. The efficiency of FITC-insulin association with the particles (%loading) at pH 7.78 and subsequent release (%release) of FITC-insulin after exposure of the insulin-particle complexes to a range of pHs were calculated from fluorescein intensity of FITC-insulin loaded into Ba salt particle. (**a**) Shows % loading of FITC-insulin into Ba salt particle (BaSO_4_, BaSO_3,_ and BaCO_3_). (**b**) Presents the formula for calculation of %loading and %release of insulin, whereas (**c**) demonstrates the data pertaining to % release of FITC-insulin at lower pHs (pH 7.14–pH 1.78).

**Figure 7 pharmaceutics-12-00710-f007:**
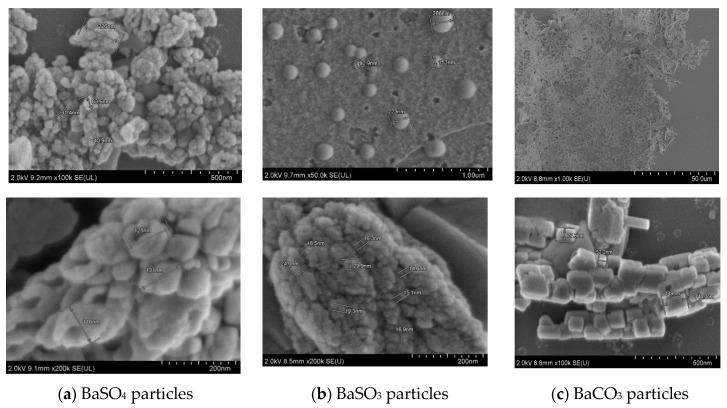
FE-SEM pictures of Ba salt particles. (**a**) BaSO_4_ particles show distinctive irregular, oval to hollow rounder shape with particle size between 30–150 nm; (**b**) BaSO_3_ particles form innate baseball shape particles of a rough surface with particle size between 15–270 nm; (**c**) BaCO_3_ particles form rod shape particles with particle size between 50–500 nm.

**Figure 8 pharmaceutics-12-00710-f008:**
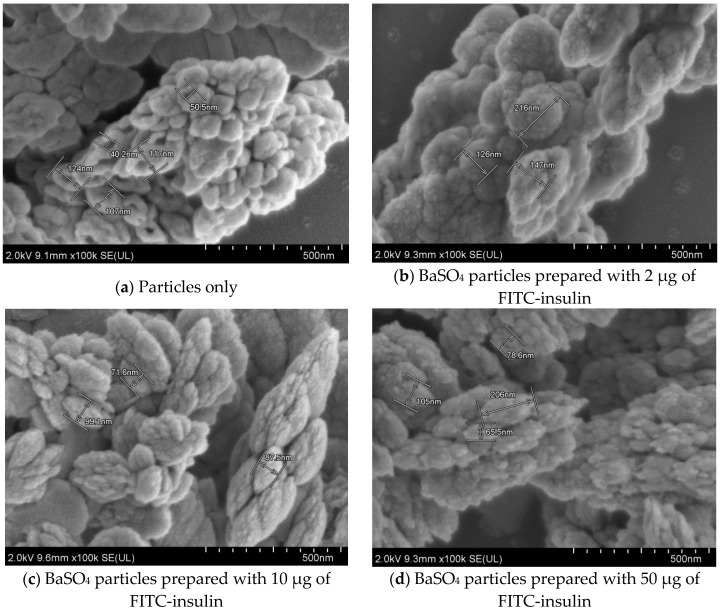
FE-SEM images of BaSO_4_ particles (free and loaded with insulin).

**Figure 9 pharmaceutics-12-00710-f009:**
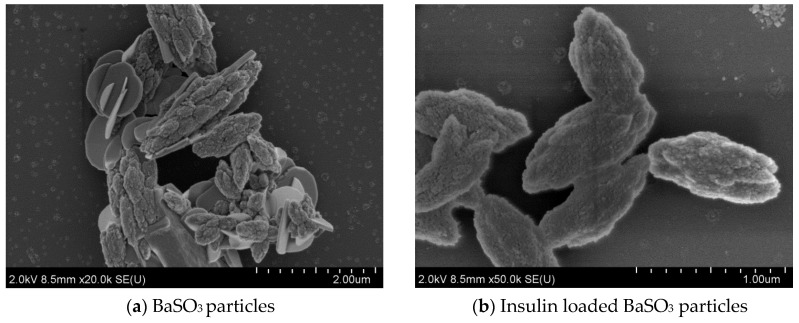
Images of empty and insulin-loaded BaSO_3_ particles. (**a**) Empty BaSO_3_ particles look more aggregated. (**b**) Insulin-loaded BaSO_3_ particles prepared with 50 µg of insulin form distinctive oval-shaped separate clusters. Pictures were taken at 2.0 kV.

**Figure 10 pharmaceutics-12-00710-f010:**
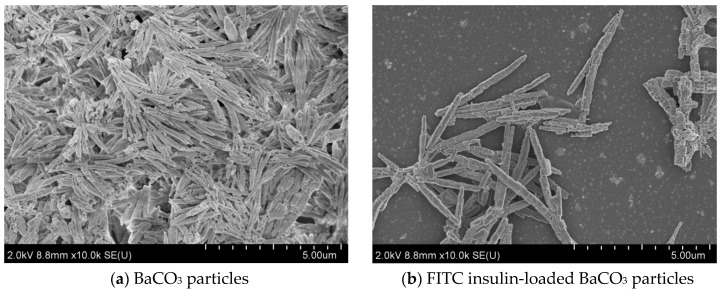
Images from free and insulin-loaded BaCO_3_ particles. (**a**) Empty BaCO_3_ particles look more aggregated. (**b**) Insulin-loaded BaCO_3_ s form distinctive rod/needle-shaped separate clusters. Pictures were taken at 2.0 kV.

**Figure 11 pharmaceutics-12-00710-f011:**
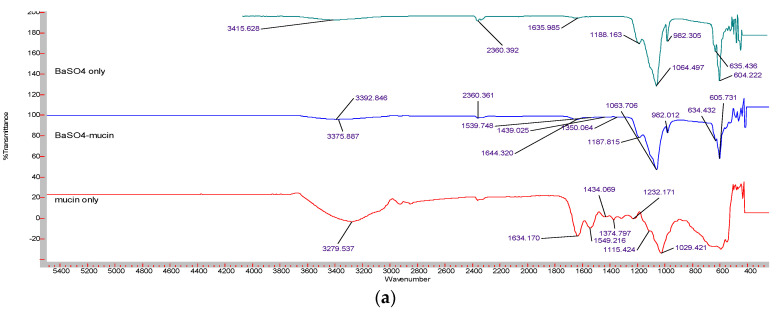
FT-IR spectrum showing mucin adhesion to three different Ba salt particles. Particle only, mucin only, and particles adhered to mucin were analyzed with FT-IR. (**a**) The spectrum for BaSO_4_ particle only, free mucin, and BaSO_4_-mucin complex. (**b**) The spectrum for BaSO_3_ particle only, free mucin, and BaSO_3_-mucin complex. (**c**) The spectrum for BaCO_3_ particle only, free mucin, and BaCO_3_-mucin complex.

**Figure 12 pharmaceutics-12-00710-f012:**
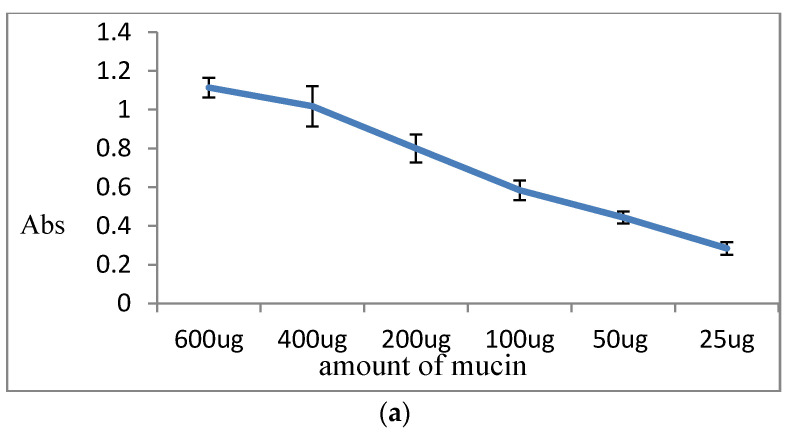
Mucin adhesion efficiency of Ba salt particles. (**a**) The standard curve with absorption vs. known amounts was obtained from solutions of different amounts of mucin (25–600 µg). (**b**) % mucin adhesion of particles was calculated from the absorbance of an unknown sample. BaSO_4_ -M: Mucin adhered to BaSO_4_ particles; BaSO_3_ -M: Mucin adhered to BaSO_3_ particles; BaCO_3_ -M: Mucin adhered to BaCO_3_ particles. Absorbance was read at 595 nm.

**Figure 13 pharmaceutics-12-00710-f013:**
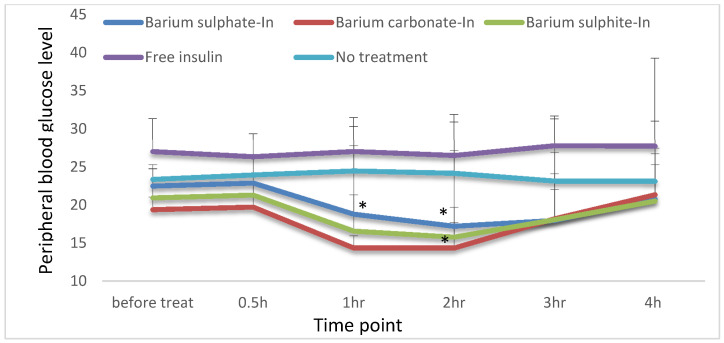
Oral delivery of free insulin and particle loaded insulin effect on hyperglycemia. The group of STZ-induced diabetic rats without any treatment was taken as a negative control, whereas the group of STZ-induced diabetic rats treated with free insulin was considered as a positive control. 3 different treatment groups of diabetic rats were orally given Insulin aspart-loaded particles of BaSO_4_, BaSO_3,_ and BaCO_3._ Each control and treatment groups have 3 animals per group. The baseline blood glucose level was read before the treatment. All 3 insulin-loaded Ba salt formulations showed a reduction in blood glucose level within 1 h of the oral administration. Mean value of (±standard deviation) was plotted for BaSO_4_, BaSO_3_, and BaCO_3_ loaded with Insulin aspart (100 IU/kg) (*p* < 0.05)**.** For BaSO_4_ particles with loaded insulin, blood glucose level was significantly low compared to the baseline level in 1–3 h (*p* < 0.05). At 4th h, the level was still low compared to the baseline level but not significant, with an ascending trend. BaCO_3_ particles with loaded insulin showed a significant decrease of blood glucose level at 1–2 h, although blood glucose level started to show a slight rise at 3rd h, and by the 4th h, it was back to the baseline level. However, insulin-loaded BaSO_3_ particles showed a reduction in blood glucose level but the reduction was not significant at 4 h.

**Figure 14 pharmaceutics-12-00710-f014:**
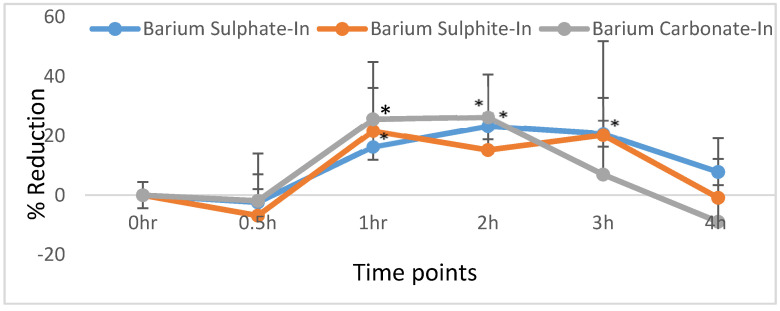
Oral administration of insulin-loaded Ba salt particles. Animals were fed with Ba salt particles with loaded insulin (100 IU/kg). Each control and treatment groups have 3 animals per group. The baseline blood glucose level was read before the treatment. % of reduction of blood glucose level at different time points. The blood glucose level at “0” h is the baseline blood glucose reading taken right before oral treatment with insulin-loaded particles. The mean value of (± standard deviation) was plotted for BaSO_4_, BaSO_3_, and BaCO_3_ loaded with Insulin aspart (100 IU/kg). A significant % reduction was observed for BaSO_4_ loaded insulin (1–3 h) and BaCO_3_ loaded insulin (1–2 h) (*p* < 0.05)**.**

**Table 1 pharmaceutics-12-00710-t001:** Formulations of three different Ba salt particles.

Salt B	Ba Salt Particles
	Salt A(1M BaCl_2,_ 5 µL)
1M Na_2_SO_4_ (2 µL)	BaSO_4_
1M Na_2_SO_3_ (2 µL)	BaSO_3_
1M Na_2_CO_3_(2 µL)	BaCO_3_
20× concentrated Ba particle formulations prepared for in vivo use	
**1M Na_2_CO_3_(40 µL)**	**BaCO_3_**
**Salt B**	**Ba salt particles**
	**Salt A (1M BaCl_2_ 100 µL)**
1M Na_2_SO_4_ (40 µL)	BaSO_4_
1M Na_2_SO_3_ (40 µL)	BaSO_3_

× all the reactants were prepared from 1M stock solution.

**Table 2 pharmaceutics-12-00710-t002:** Ba salt particle formulations loaded with insulin.

Salt B	Insulin	Salt A(1 M BaCl_2_ 5 µL)
1M Na_2_SO_4_ (2 µL)	FITC-Insulin(2 mg/mL,1–25 µL)/Insulin Aspart (1–5 IU/kg)	BaSO_4_
1M Na_2_SO_3_ (2 µL)	FITC-Insulin(2 mg/mL,1–25 µL)/Insulin Aspart (1–5 IU/kg)	BaSO_3_
1M Na_2_CO_3_(2 µL)	FITC-Insulin(2 mg/mL,1–25 µL)/Insulin Aspart (1–5 IU/kg)	BaCO_3_
20× concentrated Ba particle formulations loaded with Insulin Aspart prepared for in vivo use		
**Salt B**	**Insulin**	**Salt (1 M BaCl_2_ 100 µL)**
1M Na_2_SO_4_ (40 µL)	Insulin Aspart (100 IU/kg)	BaSO_4_
1M Na_2_SO_3_ (40 µL)	Insulin Aspart (100 IU/kg)	BaSO_3_
1M Na_2_CO_3_(40 µL)	Insulin Aspart (100 IU/kg)	BaCO_3_

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
