# Peer review of "Insulin-Loaded Barium Salt Particles Facilitate Oral Delivery of Insulin in Diabetic Rats"

_pharmaceutics, 2020, doi:10.3390/pharmaceutics12080710_

Round 1
Reviewer 1 Report
This paper seems well written, but this topic is totally out of my research area and I cannot review it scientifically.
Author Response
Reviewer 1
This paper seems well written, but this topic is totally out of my research area and I cannot review it scientifically.
Response: Thank you very much.
Reviewer 2 Report
The manuscript ‘Insulin-loaded Barium salt particles facilitate oral delivery of insulin in diabetic rats’ tried to address the problem of low bioavailability. The authors demonstrated that a novel drug delivery system (DDS) using precipitation-induced Barium (Ba) salt particles can prove efficiency oral delivery. This finding is necessary for developing oral formulations to address low bioavailability issues, so this manuscript is suitable for publication.Author Response
Reviewer 2
The manuscript ‘Insulin-loaded Barium salt particles facilitate oral delivery of insulin in diabetic rats’ tried to address the problem of low bioavailability. The authors demonstrated that a novel drug delivery system (DDS) using precipitation-induced Barium (Ba) salt particles can prove efficiency oral delivery. This finding is necessary for developing oral formulations to address low bioavailability issues, so this manuscript is suitable for publication.
Response: Thank you very much.
Reviewer 3 Report
The manuscript discusses the study and applicability of barium precipitate filled with insulin for oral administration. Several methods have been used in the studies to investigate the structure and stability of the precipitate.
My first question relates to the study of the stability of the precipitate. The particle stability assessment method is described in 2.4. point. Stability has been studied in DMEM media with different pH, but nowhere does it justify why DMEM medium was chosen for the study. To study the stability and dissolution of the formulations, pharmacopeias (e.g. Ph.Eur.) Recommend testing in an artificial stomach and intestinal fluid. What was the reason for deviating from the pharmacopeial proposal? This is not justified in the manuscript; it would be important from a pharmacist's point of view to know this. The other issue concerns the choice of DMEM. DMEM medium is used for cell culture studies, which contain important components from a cell physiological point of view (eg. amino acids, vitamins). In the present case, however, it is not clear why these components are needed to test stability?
In my opinion, it would have been more important to perform a digestion test with simulated gastric and intestinal fluids with the digestive enzyme component.
Several oral insulin formulations were mentioned in the introduction. It would have been worthwhile to compare the results of oral administration — the degree of glucose reduction — with similar results of the insulin loaded nano- and microparticles. Although the use of barium precipitate is unique, the pharmacokinetic results should have been compared with other formulation results in the discussion. This is a generally incomplete point in the manuscript.
Author Response
Reviewer 3
The manuscript discusses the study and applicability of barium precipitate filled with insulin for oral administration. Several methods have been used in the studies to investigate the structure and stability of the precipitate.
- My first question relates to the study of the stability of the precipitate. The particle stability assessment method is described in 2.4. point. Stability has been studied in DMEM media with different pH, but nowhere does it justify why DMEM medium was chosen for the study. To study the stability and dissolution of the formulations, pharmacopeias (e.g. Ph.Eur.) Recommend testing in an artificial stomach and intestinal fluid. What was the reason for deviating from the pharmacopeial proposal? This is not justified in the manuscript; it would be important from a pharmacist's point of view to know this. The other issue concerns the choice of DMEM. DMEM medium is used for cell culture studies, which contain important components from a cell physiological point of view (eg. amino acids, vitamins). In the present case, however, it is not clear why these components are needed to test stability?
Response: The particles are new and at the time of the study, their properties were completely unknown. It was important to check their responses towards pH and enzymatic activity separately as well as in combination in a simulated buffer. DMEM composition is suitable for sustenance of biological molecules including proteins. And at the same time, DMEM can accommodate pH change to a wide range. The particles (free and insulin bound) were exposed to DMEM prepared with a wide range of pHs to study their self-resistance as well as ability to protect insulin molecule against drastic pH change. Justification of method has been included in the discussion. Please refer to line-554-561, pg.21. A previous study was carried out to investigate insulin release from particles in DMEM. Protocol was adapted with modification from the following paper:
Ahmad, A; Othman, I; Zain, A.Z.M. Controlled release of insulin in blood from strontium-substituted carbonate apatite complexes. Curr Drug Deliv. 2015,12(2),210-22.
- In my opinion, it would have been more important to perform a digestion test with simulated gastric and intestinal fluids with the digestive enzyme component.
Response: Particle digestion test with simulated gastric fluid prepared with pepsin enzyme was done for a model protein, albumin and the data is shown in supplementary fig. 4.
- Several oral insulin formulations were mentioned in the introduction. It would have been worthwhile to compare the results of oral administration — the degree of glucose reduction — with similar results of the insulin loaded nano- and microparticles. Although the use of barium precipitate is unique, the pharmacokinetic results should have been compared with other formulation results in the discussion. This is a generally incomplete point in the manuscript.
Response: Relevant discussion has been added in ‘discussion’ section. Please ref. to line no. 613-654, Pg.22-23.
Reviewer 4 Report
This manuscript reports oral delivery of insulin using Ba salt formulation. Technically, this study is important and sounds interesting, and would be attractive to the researcher who is working on the relevant field. Although, overall, the study is well designed/performed, the medicinal significant of efficacy for reducing the glucose level does not seems to be quite meaningful/effective, or the presentation of the data, compare/contrast of the results with previously literature is insufficient to fathom the impact of this study. In my view, this manuscript needs major improvement in many sections including data presentation, results, and discussion, before considering for publishing for any journals. Here is some of my suggestions/comments-
- The Introduction section is decent. Compare/contrast the results (in the Discussion section) with the references mentions line 84-92. The compare/contrast should be done in terms of % of glucose reduction and time points.
- There are inconsistencies, wrong labeling/numbering in data presentation, graphs axis title, and axis scaling marks for many (if not all) of the figures, which make this manuscript difficult to follow.
- Fig.5 (section 2.4)- mention how the pH was controlled, using acid or different buffer. 1, 2, and 3 h graphs can be combined into one graph.
- Fig.6b- include FITC-insulin only sample in this experiment as a control to see how its emission influenced by pH changes.
- Fig. 13 and 14a and c can be combined to two graphs-one showing glucose level and another showing change in %. Indicate the statistical significant with respect to what samples/time points. In the caption, mention the number (n) of animal for each sample.
- Discussion section merely full of results. Include relevant discussion in the context of results, compare/contrast with the similar published works on oral delivery of insulin.
- Why would the BaSO4 salt particle bind insulin stronger than other salts? Include the possible explanation for it.
Author Response
Reviewer 4
This manuscript reports oral delivery of insulin using Ba salt formulation. Technically, this study is important and sounds interesting, and would be attractive to the researcher who is working on the relevant field. Although, overall, the study is well designed/performed, the medicinal significant of efficacy for reducing the glucose level does not seems to be quite meaningful/effective, or the presentation of the data, compare/contrast of the results with previously literature is insufficient to fathom the impact of this study. In my view, this manuscript needs major improvement in many sections including data presentation, results, and discussion, before considering for publishing for any journals.
Response: Thank you very much for your comments. Reduction of blood glucose level provides evidence of nano-insulin successfully taking a transport through GIT and crossing intestinal lining to reach blood with intact insulin.
Examples of similar in vivo test can be found in below papers:
- Wu, Z. M.; Zhou, L.; Guo, X. D.; Jiang, W.; Ling, L.; Qian, Y.; Luo, K. Q.; Zhang, L. J. HP55-coated capsule containing PLGA/RS nanoparticles for oral delivery of insulin. Int.J.Pharm. 2012, 425, 1-8.
- Pan, Y.; Li, Ying-Jian.; Zhao, Hui-Ying.; Zheng, Jun-Min.; Xu, H.; Wei, G.; Hao, Jin-Song. ; Cui, Fu-de. Bioadhesive polysaccharide in protein delivery system: chitosan nanoparticles improve the intestinal absorption of insulin in vivo. J. Pharm. 2002, 249, 139-47.
Here is some of my suggestions/comments-
1. The Introduction section is decent. Compare/contrast the results (in the Discussion section) with the references mentions line 84-92. The compare/contrast should be done in terms of % of glucose reduction and time points.
Response: Discussion based on % of glucose reduction and time points has been added. Please ref. to line no. 613-654, Pg.22-23.
2. There are inconsistencies, wrong labeling/numbering in data presentation, graphs axis title, and axis scaling marks for many (if not all) of the figures, which make this manuscript difficult to follow.
Response: Corrected: fig.5 axis mark, fig.8 legend,
3. Fig.5 (section 2.4)- mention how the pH was controlled, using acid or different buffer. 1, 2, and 3 h graphs can be combined into one graph.
Response: In Fig.5 (section 2.4), pH was measured immediately after adding DMEM media prepared with different pHs (pH7.78-pH1.8). Figures are divided into 1,2 and 3h graphs for clear presentation, avoiding confusion with so much data crumbled in one graph.
4. Fig.6b- include FITC-insulin only sample in this experiment as a control to see how its emission influenced by pH changes.
Response: FITC-Insulin only sample was used to prepare the standard curve which has been included in supplementary figure 1 for the record.
5. Fig. 13 and 14a and c can be combined to two graphs-one showing glucose level and another showing change in %. Indicate the statistical significant with respect to what samples/time points. In the caption, mention the number (n) of animal for each sample.
Response: Fig. 13 and Fig. 14 have been re-arranged according to the reviewer’s comment. Statistically significant points and animal number have been mentioned in the caption.
6. Discussion section merely full of results. Include relevant discussion in the context of results, compare/contrast with the similar published works on oral delivery of insulin.
Response: We have added the relevant paragraph in discussion. Please ref. to line no. 613-654, Pg.22-23.
7. Why would the BaSO4 salt particle bind insulin stronger than other salts? Include the possible explanation for it.
Response: According to our observation, in vitro insulin loading efficiency was similar for all 3 different types of Ba salt particles, with BaCO3 particles demonstrating slightly higher binding affinity for insulin (100%) than BaSO4 and BaSO3 particles; however, in vitro stability and protection from pH and enzymatic activity data showed stronger protection from BaSO4 and BaCO3 particles compared to BaSO3 particle. This could be owing to the difference in crystallinity and/or self-aggregation properties of those particles. We have added this explanation in page 21, line 590-593).
Round 2
Reviewer 3 Report
The Authors answered my previous questions, so I accept the manuscript in present form.
Author Response
Reviewer 3
Comment (Second round): The Authors answered my previous questions, so I accept the manuscript in present form.
Response: Thank you very much.
Reviewer 4 Report
In many places formula of Ba salts are written without subscripting the numbers. Many graphs still do not have axis with proper marking. My suggestion would be publish this manuscript after taking care of formatting/presentation issues.
Author Response
Reviewer 4
Comment (Second round): In many places formula of Ba salts are written without subscripting the numbers. Many graphs still do not have axis with proper marking. My suggestion would be publish this manuscript after taking care of formatting/presentation issues.
Response: Thank you for the comment.
- Axis labelling has been corrected for Fig. 5, Fig. 6(c) and supplementary Fig. 3.
- Ba salts have been written (corrected) with subscript in Fig. 5, Fig. 6 and supplementary Fig.4(b).